# The role of cryptic ancestral symmetry in histone folding mechanisms across Eukarya and Archaea

Haiqing Zhao[1,3¤*], Hao Wu[1], Alex Guseman[2], Dulith Abeykoon[2], Christina M. Camara[2], Yamini Dalal[3*], David Fushman[1,2*], Garegin A. Papoian[1,2*]

**1** Biophysics Program, Institute for Physical Science and Technology, University of Maryland, College Park, Maryland, United States of America, **2** Department of Chemistry and Biochemistry, University of Maryland, College Park, Maryland, United States of America, **3** Laboratory of Receptor Biology and Gene Expression, National Cancer Institute, National Institutes of Health, Bethesda, Maryland, United States of America

¤ Current address: Department of Systems Biology, Columbia University, New York, New York, United States of America

* hz2592@columbia.edu (HZ); dalaly@mail.nih.gov (YD); fushman@umd.edu (DF); gpapoian@umd.edu (GAP)

**Data Availability Statement:** The molecular dynamics simulation data, model setup, used codes and scripts are available on zenodo (https://zenodo.org/records/8226152). The used AWSEM

## Abstract

Histones compact and store DNA in both Eukarya and Archaea, forming heterodimers in Eukarya and homodimers in Archaea. Despite this, the folding mechanism of histones across species remains unclear. Our study addresses this gap by investigating 11 types of histone and histone-like proteins across humans, Drosophila, and Archaea through multiscale molecular dynamics (MD) simulations, complemented by NMR and circular dichroism experiments. We confirm and elaborate on the widely applied "folding upon binding" mechanism of histone dimeric proteins and report a new alternative conformation, namely, the inverted non-native dimer, which may be a thermodynamically metastable configuration. Protein sequence analysis indicated that the inverted conformation arises from the hidden ancestral head-tail sequence symmetry underlying all histone proteins, which is congruent with the previously proposed histone evolution hypotheses. Finally, to explore the potential formations of homodimers in Eukarya, we utilized MD-based AWSEM and AI-based AlphaFold-Multimer models to predict their structures and conducted extensive all-atom MD simulations to examine their respective structural stabilities. Our results suggest that eukaryotic histones may also form stable homodimers, whereas their disordered tails bring significant structural asymmetry and tip the balance towards the formation of commonly observed heterotypic dimers.

## Author summary

Histones are among the most conserved proteins but diverge into different oligomers with different lengths of tails in Eukarya and Archaea. This work explores the general folding mechanism of histone dimers, their unique dynamics featuring different formats and possible causes from molecular evolution. Through multiscale molecular dynamics

software can be downloaded from GitHub (https://github.com/adavtyan/awsemmd).

**Funding:** This work is supported by NCI-UMD partnership program for Integrative Cancer Research and the Ann G. Wylie Dissertation Fellowship (H.Z.), the Amazon Web Services Artificial Intelligence Award (G.P.), the NIH Intramural Research Program (Y.D.), and the NIH grant GM065334 (D.F.). The funders had no role in study design, data collection and analysis, decision to publish, or preparation of the manuscript.

**Competing interests:** The authors have declared that no competing interests exist.

simulations and NMR/CD experiments, the folding-upon-binding mechanism was confirmed in 11 histone and histone-like protein types, including histone variants such as CENP-A and two transcription factors—one from Drosophila and another from humans. This study suggests an interesting folding pathway, and a stable non-native dimer conformation which has an inverted geometry. The latter reveals a hidden head-tail sequence symmetry of hydrophobic residues that lines behind all histone proteins, aligning with a previously proposed hypothesis on histone evolution. Based on this symmetry, the structure of eukaryotic histone homodimers was predicted using AWSEM model and Alpha-fold-Multimer and examined through atomistic simulations. Lastly, we particularly investigated the role of histone tails, unique to eukaryotes, in the folding of histone hetero-dimers and homodimers, shedding light on their evolutionary significance.

## Introduction

In Eukarya, histones are fundamental proteins for chromosome packaging. In nucleosome, the basic unit of chromosome, histones are assembled as four dimers, two H3/H4 heterodimers forming a tetramer and two H2A/H2B heterodimers separately [1]. Besides the canonical histones composing the majority of eukaryotic nucleosomes, histone variants and other histone-like proteins carry out specific functions in the nucleus [2, 3]. In Archaea, histones are encoded to package and compact DNA into continuous hypernucleosomes [4, 5]. Among many different histone oligomerization variations, dimer is the smallest unit reported in both Eukarya and Archaea, with regard to either structure or function. Nevertheless, in Eukarya histones exist as hetero-dimers with long terminal tails (Fig 1A), while in Archaea both homo- and hetero-dimer exist with no histone tails (Fig 1B).

Despite their function and sequence diversities across species, histones and histone-like proteins possess the same structural motif, known as histone-fold, which is comprised of three alpha helices connected by two loops [6]. Two histone-fold monomers assemble into a "handshake" pattern forming a dimer in an intertwined, head-to-tail manner [7] (Fig 1A and 1B). A scaling analysis of the radius of gyration ($R_g$) as a function of the protein size shows that histone dimer fits almost perfectly to the empirical scaling relation of monomeric proteins (Fig 1C, and more details in S1 Text), suggesting that histones may form obligatory dimers. Indeed, experimentally, it was known that eukaryotic histones fold and form complexes only in presence of their binding partner [8, 9]. The stability of H2A/H2B dimers and (H3/H4)$_2$ tetramers were studied by a series of denaturation experiments [10–13]. Karantza *et al.* reported that during the unfolding of either H2A/H2B or (H3/H4)$_2$, individual folded monomers were not detectable, indicating a direct transition from one folded dimer to two unfolded monomers [10].

However, the molecular principles underlying this cooperative behavior have not been elucidated. In addition, whether or not these principles vary among eukaryotic histones, archaeal histones, and other histone-like structures remain unknown. Deeper insights into the folding dynamics of histone and histone-like proteins can not only shed light on their evolutionary basis and divergence, but also help understand their higher level structural organizations such as tetramer or octamer formation [14, 15], nucleosome interactome [16, 17], and chromosome organization [18–22]. In this work, we aim to build mechanistic understanding of histone folding at molecular detail and more broadly, to compare the folding mechanisms of histones across Eukarya and Archaea.

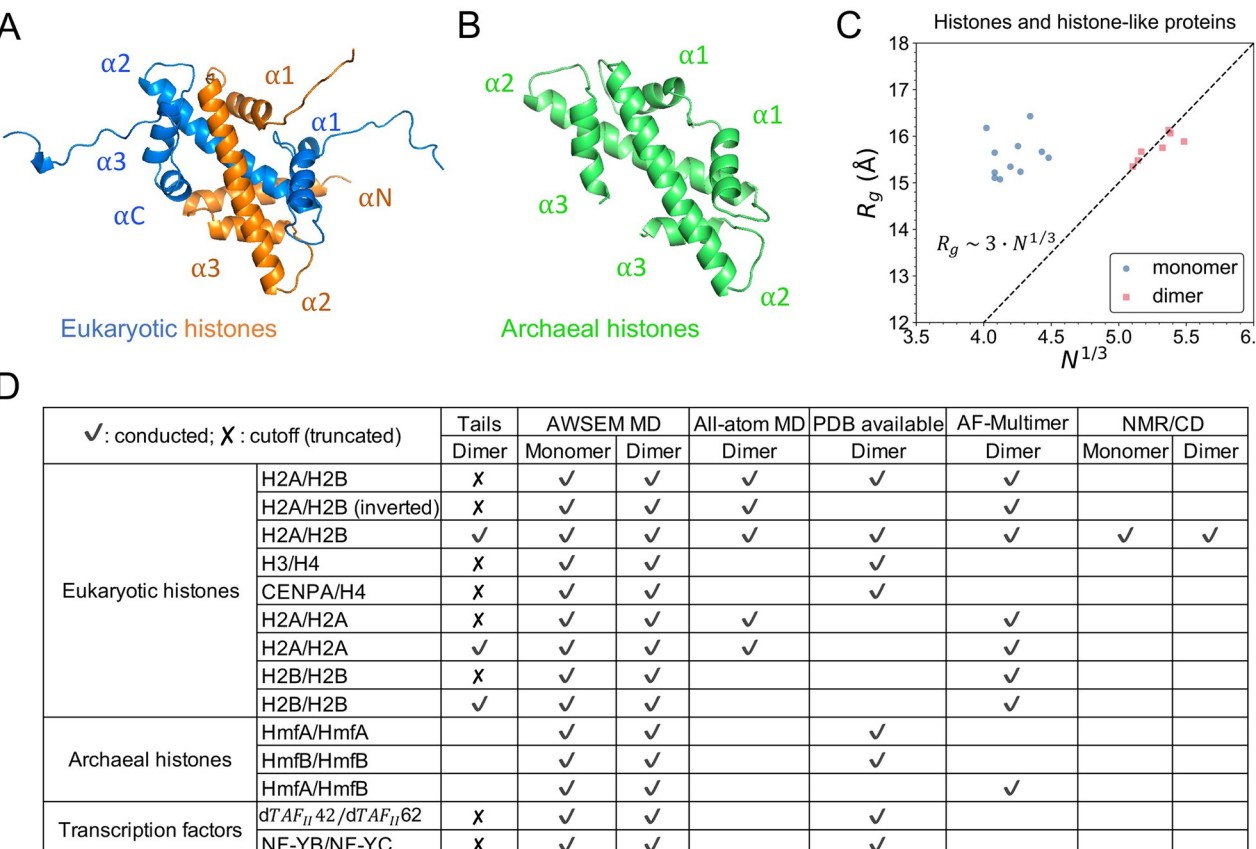

**Fig 1. Structures of eukaryotic and archaeal histones and a summary table of studied systems.** (A) The scaffold of eukaryotic histone dimer H2A/H2B (blue/orange, PDB: 1AOI) consists of three $\alpha$ helices in each protein, plus two terminal helices and long histone tails. (B) The structure of homotypic archaeal histone dimer (HmfB)$_2$ (green, PDB: 1A7W) has three $\alpha$ helices each with no histone tails. (C) Polymer scaling fitting of the $R_g$ and sequence length suggests that histone dimers act more as "monomeric" proteins. 7 histone(-like) dimer structures from PDB and their 11 monomers are included in this plot. The dashed line is the empirical relation of $R_g$ and $N$ from a survey study of 403 globular monomeric proteins [25]. (D) This table summarizes the 13 types of heterotypic and homotypic histone(-like) dimers that are studied in this work by different methodologies. A blank cell in Tails/PDB column means "non-existent" while in other columns it means "not conducted".

To address these questions, we first employed molecular dynamics (MD) simulations based on a coarse-grained protein force field AWSEM [23]. AWSEM, the associative memory, water mediated, structure and energy model, is a transferrable coarse-grained protein model deeply based on the funneled free energy landscape of protein folding. Primarily employing the simulated annealing, AWSEM has undergone extensive tests for its satisfactory capability in predicting protein structure and protein-protein interface [23, 24]. Here, we use AWSEM to study protein foldings of seven different histones and four histone-like transcription factors (Fig 1D). We find that all studied histones or histone-like proteins are unstable as monomers. In agreement with the abovementioned prior experiments, our simulations show that two histone monomers cooperatively fold into a stable complex, revealing folding-upon-binding dynamics. We complemented our computational predictions by experimental investigations using Nuclear Magnetic Resonance (NMR) spectroscopy and circular dichroism (CD). The experimental data indicate that histones adopt an ordered structure only upon binding their partners at equimolar stoichiometry. All studied systems and used methodologies are summarized in Fig 1D.

Subsequent free energy (FE) calculations revealed an unexpected inverted non-native conformation of the histone dimer which occupies an FE minimum. The energy barrier between the native and non-native states was estimated around 8–9 kcal/mol. Further analysis of protein sequence alignment shows that our observation of non-native conformation is consistent with the well-known evolution hypotheses of histone-fold structural motif [26], unveiling a hidden sequence symmetry in contemporary histones. Moreover, our simulations suggest that archaeal histones and other histone-like proteins including transcription factors also exhibit folding upon binding, with the abovementioned non-native dimer conformation potentially being a low-energy state.

Lastly, we applied the AWSEM model as well as the deep-learning structure prediction algorithm AlphaFold-Multimer (AF-Multimer) [27, 28] to predict the structure of hypothetical eukaryotic histone homodimer and the potential effects of histone tails. Structures obtained from these two independent methodologies were examined through extensive all-atom MD simulations. Together, these simulations suggest the important role of histone tails in determining the folding and formations of various eukaryotic histone dimers, which distinguishes eukaryotic histones from archaeal ones.

## Methods

### AWSEM and all-atom MD simulations

The coarse-grained MD simulations were carried out using LAMMPS with the AWSEM model, under non-periodic shrink-wrapped boundary condition and Nose-Hoover thermostat control. The Hamiltonian of AWSEM contains both physical interaction terms and a bioinformatics-inspired memory term, $V_{AWSEM} = V_{backbone} + V_{contact} + V_{burial} + V_{H-bond} + V_{FM}$. Details of every term are described in Davtyan *et al.* [23] and Papoian *et al.* [29]. AWSEM model has been applied to predict protein structures and protein-protein interactions [23, 24], and in particular to histone dynamics such as chaperone-assisted histone dimer [30], tetramer and octamer [14], histone tails [31], nucleosome [32] and linker histone [33]. In this work, the fragment library used in the last term is composed of local fragments (3–9 residues long) either from the structure of histone monomers for the dimer folding study, or called "homologs-allowed", from the structural fragments of their homologous sequences with less than 95% sequence identity to the target. The latter was used to explore possible monomer and homodimer structures. The control group of heterodimer in the homodimer study used the same setup.

To find low-energy conformations, simulated annealing was carried out by decreasing the simulation temperature from 600 K to 200 K. The initial conformations of the unfolded state were prepared at 1000 K. The final conformation that has the lowest energy was chosen as the prediction outcome. Ranging from 0 to 1, the order parameter $Q$ (defined as $\frac{1}{N}\sum_{i<j-2}exp\left[-\frac{(r_{ij}-r_{ij}^N)^2}{2\sigma_{ij}}\right]$, see Section S1.3 in S1 Text for details) was used to quantify the structural similarity to the native structure. To compute free energy profiles at a certain temperature, umbrella sampling was applied with $Q$ as the reaction coordinate. WHAM [34] was used to remove the potential bias. To mimic a finite protein concentration, a weak distance constraint via a harmonic potential was applied between the centers of mass of two monomers (the spring constant $k = 0.02$ kcal/mol/Å$^2$). Protein sequences of all studied systems are provided in S1 Fig. A detailed setup of the AWSEM model and its complete simulation results are in Section S1.1 in S1 Text.

The all-atom MD simulations were performed using OpenMM 7.6.0 [35] on GPUs, with the Amber ff14SB protein model and the TIP3P water model. Initial conformations were either

from the crystal structure, or predictions of AWSEM-MD and AF-Multimer. All the simulated systems were solvated in a 150 mM KCl solution. In total, eight molecular systems were separately simulated with two independent replicas for each. After standard energy minimization and equilibration, 800 ns were run for each replica, in total giving 12.8 microseconds of simulations. Results in the main context are from one replica. A detailed setup of the all-atom MD simulations and its complete results are in Section S1.2 in S1 Text.

## AlphaFold-multimer and multiple sequence alignment

The AlphaFold-Multimer predictions for protein-protein complex were carried out using ColabFold [36] on cloud service platform Google Colaboratory, with the fast homology search method MMseqs2 [37], the paired alignment and amber relaxation options.

All used protein sequences in this study were downloaded from UniProtKB database [38] and aligned using MUSCLE v3.8.31 [39]. The generated multiple sequence alignments were visualized by Multiple Align Show, https://www.bioinformatics.org/sms/multi_align.html. Structure and simulation conformation figures used in this work are generated by PyMOL Molecular Graphics System Version.

## NMR and CD experiments

Unlabeled and $^{15}$N labeled histones human H2A (type 2-A) and H2B (type 1-C) were expressed in *E. coli* and purified from inclusion bodies using cation exchange chromatography. Their correct mass was confirmed by mass spectrometry. All NMR experiments were performed at 23°C on a 600 MHz Bruker Avance-III NMR spectrometer equipped with TCI cryoprobe. Proteins were dissolved at 100–200 $\mu$M in 20 mM sodium phosphate buffer (pH 6.8) containing 7% D$_2$O and 0.02% NaN$_3$. NMR data were processed using TopSpin (Bruker Inc.)

Circular dichroism (CD) spectra of histone proteins at 0.40 mg/mL concentration in 20 mM sodium phosphate buffer at pH 6.8 were acquired on a Jasco J810 Spectro-Polarimeter using a Peltier-based temperature-controlled chamber at 25°C and a scanning speed of 50 nm/min. A quartz cell (1.0 mm path length) was used. All measurements were performed in triplicate. To determine the secondary structure content, the CD data were analyzed using the DichroWeb server [40]. Two methods were used in parallel: (i) CDSSTR, a singular value decomposition (SVD)-based approach employing two datasets (7 and SMP180) from the DichroWeb server and (ii) K$_2$D, a neural network-based algorithm trained using reference CD data [41].

## Results

### The Folding-upon-binding mechanism of histones

The individual folded monomers were not detectable in the past unfolding experiments of histone dimers or tetramers [10, 11]. Our first aim was to investigate the molecular basis for this observation using AWSEM-MD. For that, we simulated 11 different histone monomers which include the eukaryotic canonical histones such as H2A and H2B, the variant histone CENP-A, and archaeal histones HmfA and HmfB. Histone tails were not included. Ten independent temperature annealing simulations were run for each monomer. Our simulation indicates that histone monomers are highly mobile at the tertiary structure level, as evidenced by the structural similarity measurement $Q$ values less than 0.4 (Fig 2A and S2 Fig, definition of $Q$ see Methods), indicative of far-from-native conformations. The corresponding root-mean-square deviations (RMSD) from the native structure are more than 8 Å. Clustering analysis of the final conformations from different runs exhibits that no consensus tertiary structure was

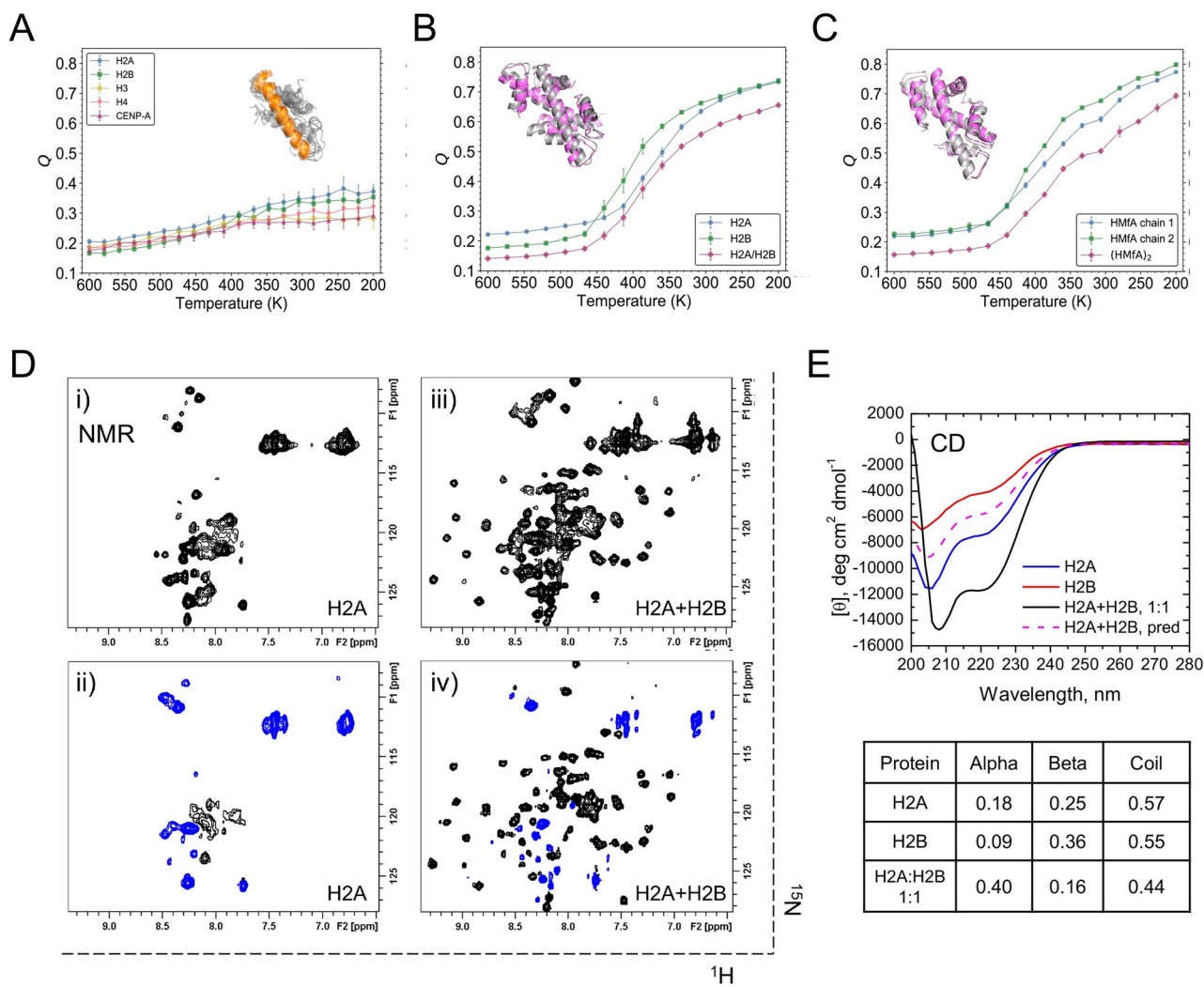

**Fig 2. Histone heterodimers fold in MD simulations and NMR experiments, not monomers.** (A) *Q* values as a function of the annealing temperature are plotted for simulated histone monomers, with the mean and standard deviation displayed as circles and error bars. Aligned final snapshots of H2A by $\alpha 2$ helix (orange) show that no stable tetiary structure formed. (B-C) *Q* analysis shows that histone monomers (blue, green) cooperatively fold and bind into a dimer (magenta). Two example systems are shown, human H2A/H2B (B) and archaeal histones $(HMfA)_2$ (C). The final conformations (magenta) are well aligned with their native states (gray). (D-E) NMR and CD studies of H2A and H2B upon their complex formation. (D) $^1H$-$^{15}N$ NMR spectra of $^{15}N$-labeled H2A alone (i) and in the presence of unlabeled H2B at a 1:1 molar ratio (iii). Heteronuclear steady-state $^{15}N\{^1H\}$ NOE spectra with amide proton presaturation recorded for $^{15}N$-labeled H2A alone (ii) and in the presence of unlabeled H2B at an equimolar ratio (iv). In these spectra, contours with positive intensities are colored black while negative intensities are blue. (E) CD spectra of H2A (blue) and H2B (red) alone and in an equimolar mixture (black). The total concentrations of the proteins are the same in all three cases. Also shown (dashed magenta) is the expected CD spectrum of the equimolar mixture if no structural changes in either protein. The table shows the secondary structure composition of the proteins estimated from these experimental CD data using $K_2D$ algorithm.

found (S3 Fig). On the other hand, histone monomers exhibit stable secondary structure elements, especially the long $\alpha 2$ helix, which are similar to the native structure (Fig 2A). Without the stabilizing contacts from another monomer, the partially formed two short helices $\alpha 1$ and $\alpha 3$ of one histone are highly mobile, moving around the $\alpha 2$ helix, leading to significant tertiary disorder.

We next applied a similar annealing protocol to investigate the folding of histones in the presence of their cognate binding partner. We calculated *Q* values of the entire dimer and of

its component monomers. It shows that during the annealing run, $Q_{dimer}$ increases roughly concurrently with $Q_{monomer}$ (Fig 2B and 2C), indicating a clear structural transition wherein the two interacting monomeric chains cooperatively fold and bind. More than 95% native contacts were finally correctly predicted in the absence of any bias towards these contacts by the AWSEM force field (S4 Fig). Besides H2A/H2B, analogous studies were undertaken for human histones H3/H4 and histone variant CENP-A/H4 (S5 Fig), archaeal histones (HMfA)$_2$ (Fig 2C) and (HMfB)$_2$ (S5 Fig), Drosophila transcription factors dTAF$_{II}$42/dTAF$_{II}$62 and human transcription factor NF-YB/NF-YC (S5 Fig). It is noteworthy that we used simulated annealing procedure to predict the heterodimer structure of archaeal histones HMfA/HMfB, which was experimentally studied [42] but structurally not resolved (S6 Fig). All the results suggest that histones and histone-like transcription factors only fold upon binding with their partners. Interestingly, from the folding trajectories of the simulated histone dimers, it seems that histones first fold their long $\alpha$2 helices, then form other $\alpha$ helices and higher-level intermolecular tertiary structures (see S1 Video).

To experimentally probe histone dimer folding, we carried out NMR and CD measurements on H2A and H2B. $^1$H-$^{15}$N NMR spectrum of H2A alone (Fig 2D i) shows a narrow spread of NMR signals resulting in signal crowding in the region typical for amide signals of unstructured/unfolded proteins. The negative or close to zero signal intensities observed in the heteronuclear steady-state NOE spectrum of $^{15}$N-labeled H2A recorded upon pre-saturation of amide protons (Fig 2D ii) are a clear indication that the protein is unstructured and highly flexible. Upon addition of unlabeled H2B, we observed a dramatic change in the $^1$H-$^{15}$N NMR spectra of $^{15}$N-labeled H2A, wherein new signals (corresponding to the bound state) emerge and increase in intensity until they saturate at ca. equimolar H2B:H2A ratio (Fig 2D iii). Concomitantly, the unbound signals reduce in intensity and practically disappear at the saturation point. This behavior of the NMR signals, which exhibit essentially no gradual shifts, indicates that the binding is in slow exchange regime on the NMR chemical shift time scale. In contrast to the unbound state, the signals of $^{15}$N-labeled H2A in complex with H2B (Fig 2D iv) show a significant spread, indicating that the bound state of H2A is well structured. Also, many H2A signals in the heteronuclear steady-state NOE spectra recorded at these conditions have positive intensities, which are characteristic of a well-folded state of the protein [43]. A similar behavior was observed for $^{15}$N-labeled H2B, which is unstructured in the unbound state and folds upon complex formation with H2A (S7 Fig).

The NMR data presented above suggested that only with each other can H2A and H2B fold into a histone dimer with well-defined structures. To extend this analysis further, we performed CD measurements, which allow one to assess the helical content of a protein experimentally. The CD results demonstrate a significant increase in the helical content of these proteins upon formation of the H2A/H2B heterodimer (Fig 2E). Together, these experimental results indicate that in isolation H2A and H2B are disordered but adopt a well-defined tertiary structure upon binding to each other, which is consistent with a previous experimental study of H2A/H2B's thermodynamic stability [10]. The 40% helical content of the human H2A/H2B heterodimer observed here is in excellent agreement with that for *Xenopus laevis* H2A/H2B [44]. Overall, our experiments and the above elaborated simulations are in qualitative agreement.

## Free energy landscape highlights a stable inverted histone dimer

Besides the native conformation, we observed an interesting non-native structure in the annealing simulations of core histone dimers (Fig 3). Compared to that of the native complex, the secondary structural elements of the non-native complex show very little differences

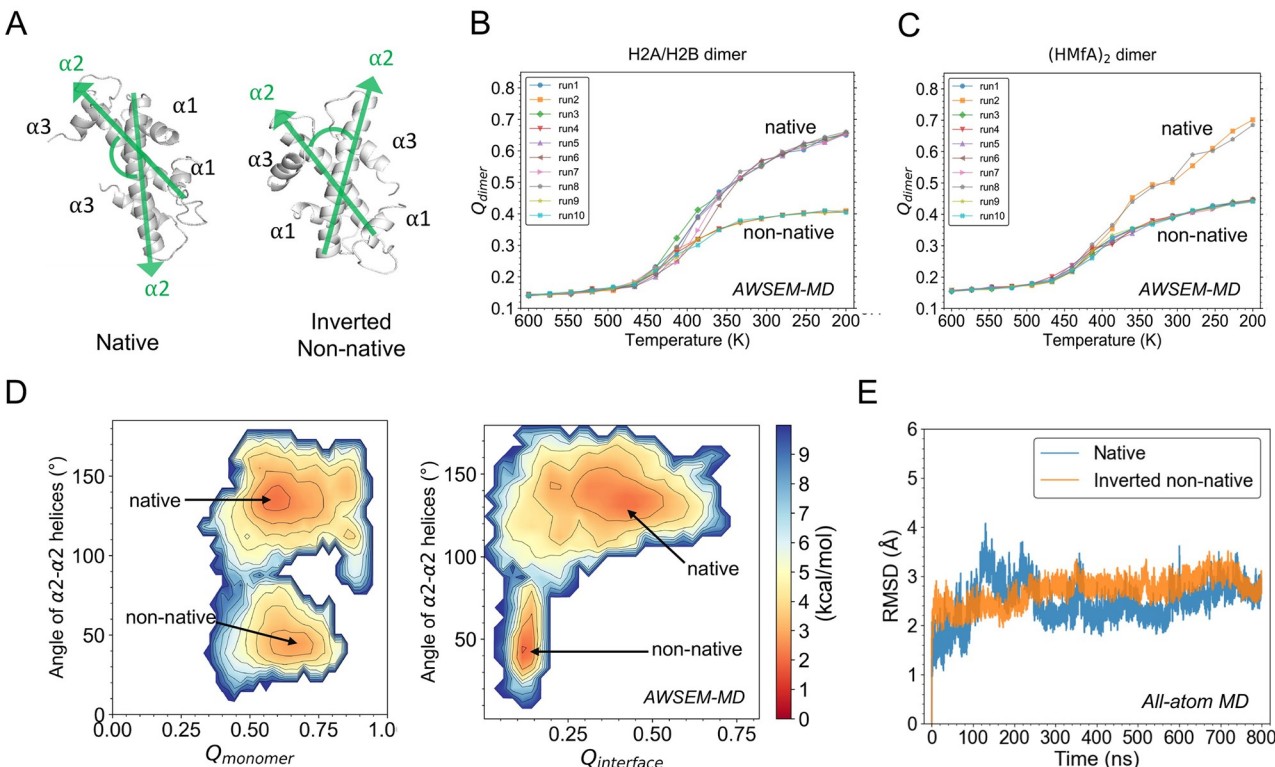

**Fig 3. Inverted non-native conformation is found to be a stable formation of histone dimer.** (A) The native and non-native conformation of H2A/H2B found in AWSEM simulations are shown. Their major difference is measured by the angle between the $\alpha2$-$\alpha2$ helices (green) with each vector arrow pointing from N- to C-terminal. (B-C) $Q_{dimer}$ analysis shows that both native and non-native conformations are observed in the simulated annealing runs of eukaryotic H2AH2B (B) and the archaeal dimer (HMfA)$_2$ (C). (D) Free energy profiles of H2A/H2B are projected on $Q_{monomer}$ and the angle of $\alpha2$-$\alpha2$ helices (left), and on $Q_{interface}$ and the $\alpha2$-$\alpha2$ helices angle (right). Two energy minima are found, corresponding to the native-like and inverted non-native conformation respectively. (E) All-atom MD simulations show comparable stability of inverted non-native conformation to that of native structure of histone dimer.

but are arranged differently, leading to an inverted inter-molecular tertiary structure. The $\alpha1$ helix of one histone is in proximity of the other's $\alpha3$ helix, instead of two $\alpha1$ helices close to each other as in the native structure (Fig 3A). If we define a direction pointing from N- to C-terminus for each histone, the angle between the two $\alpha2$ helices in the inverted conformation is nearly complementary to that of the native. Not only in eukaryotic histones (Fig 3B), this inverted conformation was also found in other simulated histone-fold systems, including archaeal histones (Fig 3C) and transcription factors dTAF$_{II}$42/dTAF$_{II}$62 dimer and NF-YB/ NF-YC dimer (S8 Fig; also see the S1 Video for dynamics details). Interestingly, the native and inverted complexes are energetically comparable according to the AWSEM potential (S9 Fig).

To have a quantitative understanding of histone complex folding, we probed the FE landscape of H2A/H2B using umbrella sampling with AWSEM. The following three reaction coordinates were used to project the obtained FE profiles: the angle between two $\alpha2$ helices to describe the general inter-chain geometry (Fig 3A), $Q_{monomer}$ to show how well histone monomer is folded, and $Q_{interface}$ to show the nativeness of formed histone dimer interface. In both of the FE landscapes (Fig 3D), there are two major basins. The first basin (top one in both panels) is located at the $\alpha2$-$\alpha2$ angle of 140°, both $Q_{monomer}$ and $Q_{dimer}$ high, all of which are consistent with the native conformation. The second basin (lower one in the figure) has

the $\alpha2$-$\alpha2$ angle of 40˚, high $Q_{monomer}$ but low $Q_{dimer}$, indicating the formation of native-like monomers and a "mismatched" binding interface. The representative structures at the two basins show that they are the native-like and the inverted non-native conformations as we found in previous annealing runs. The native basin seems to be significantly broader, suggesting entropic stabilization. The free energy barrier between the two states is relatively high at ∼8–9 kcal/mol.

To further test the structural stability of the newly-found non-native histone dimer, we also took the final snapshot of its inverted conformation from AWSEM prediction and used that as the initial conformation to perform extensive all-atom MD simulations in explicit solvent (see Methods and Section S1.2 in S1 Text for technical details). The subsequent RMSD analysis indicates that the non-native structure reached a steady state and maintained remarkable structural stability (Fig 3E), which is comparable to the simulation started from the native structure. More result analyses are in S10 Fig.

## Sequence symmetry explains the inverted conformation

Next, we explored the molecular interactions that potentially promote the non-native histone dimeric complex formation. It is known that in the native complex, $\alpha1$, $\alpha2$, and $\alpha3$ helices of one histone interact with the other histone through hydrophobic interactions (Fig 4A). These hydrophobic residues are typically conserved among all histone structures (S1 Fig). Interestingly, we found that in the inverted non-native complex these interactions are still favored when swapping $\alpha1$ and $\alpha3$ helices. To further explain its biological basis, we reversed the histone sequences from C- to N-terminus, and carried out sequence alignments between the reversed and normal order histone sequences (Fig 4B). Consequently, the multiple sequence

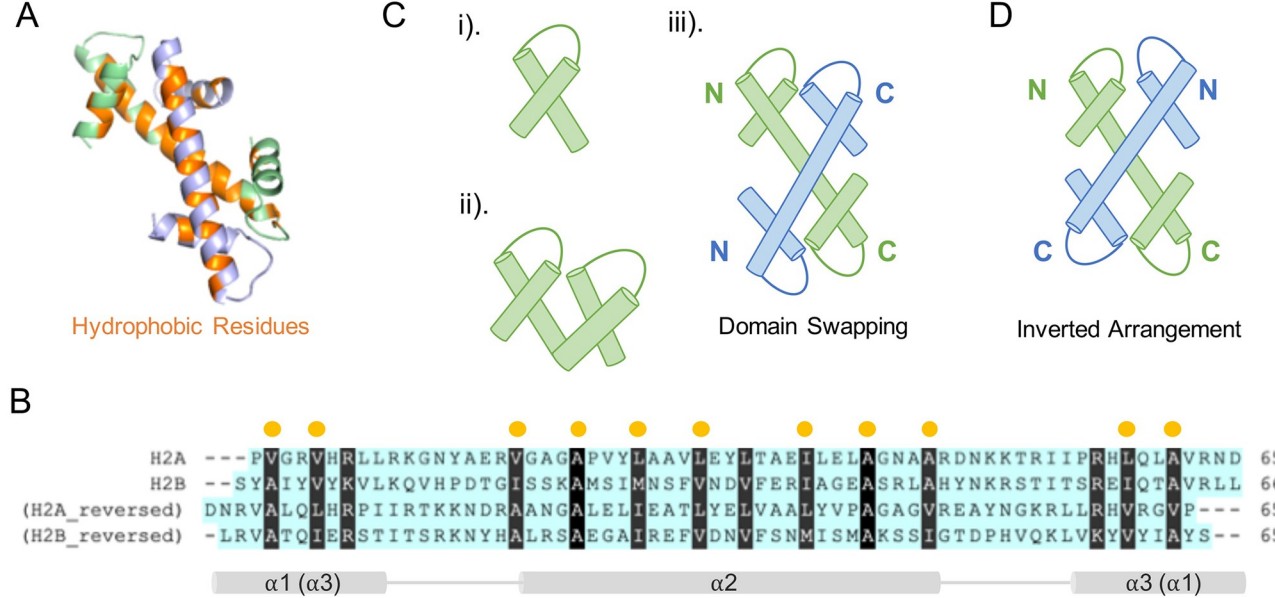

**Fig 4. Sequence symmetry of hydrophobic residues explains the predictability of inverted histone-fold structure.** (A) Hydrophobic interactions (orange) dominate the formation of binding interface of H2A/H2B (colored in green/purple). (B) Protein sequence alignments of histones H2A, H2B and their reversed sequences highlight the symmetrical distribution of hydrophobic residues. Hydrophobic residues are particularly marked. (C) Cartoon schemes illustrate previously proposed histone evolution hypotheses: i) histones may originate from one single helix-strand-helix structural motif HSH; ii) two HSH patterns form one monomer through duplication, differentiation and fusion; iii) domain swapping between two monomers (colored in green and blue) forms a histone-fold structure. (D) The inverted arrangement based on hydrophobic interactions could be an alternative formation of histone-fold structure.

alignments reveal a symmetric distribution of hydrophobic residues among histones and their head-tail reversed sequences. Thus, one may expect that in an N-C-terminal inverted arrangement, histones should still be able to maintain the required hydrophobic interactions for a stable formation due to the conserved symmetry of hydrophobic positions.

The finding of head-tail sequence symmetry and inverted histone dimer well support the previously proposed histone evolution hypotheses. The latter posits that histones may have arisen through the duplication and differentiation of a primordial helix-strand-helix motif [7, 45] (Fig 4C i-ii), and domain swapping may have triggered the subsequent dimerization of two histone monomers [26, 46] (Fig 4C iii). Thus, it is possible that the conservation of hydrophobic positions found in histone sequences, no matter in a normal or reversed order, originated from the same ancestral peptide. Meanwhile, the conserved hydrophobicity in the four segments of two histones explains the rationale of two alternative ways of the crossing forming the hand-shake motif for the characteristic of the histone fold.

## AWSEM-MD and AlphaFold-multimer predict eukaryotic homodimer structure

The above observed inverted non-native histone-fold conformation was consistently observed in archaeal histones $(HmfA)_2$, $(HmfB)_2$, HmfA/HmfB, and transcription factors $dTAF_{II}42/dTAF_{II}62$ dimer and NF-YB/NF-YC dimer (S8 Fig). These results indicate a well-conserved folding mechnism of histone-fold structures regardless of species and function distinctions, which prompted us to investigate the hypothetical eukaryotic homodimeric histones. As known, a significant difference of histone oligomers between Eukarya and Archaea is that eukaryotic histones naturally exist as heterodimer while in Archaea both homodimer and heterodimer histones are prevalent. In early biochemical investigations of histone-histone interactions, sedimentation experiments showed that eukaryotic homotypic histones may exist at high histone and salt concentrations [47, 48], however, without additional structural elaboration.

Among the predicted structures, we found both native-like and the above mentioned inverted conformations. The RMSD between native-oriented H2A/H2A complex and the native H2A/H2B structure is 6.0 Å. We then applied AlphaFold-Multimer [27, 28] in Colab-Fold [36] to carry out analogous complex structure prediction for H2A/H2A. Interestingly, both native-oriented and inverted structures were predicted there as well, consistent with our predictions from using AWSEM model. The top-scored prediction from AF2 is a native-like histone-fold structure, with an RMSD of 2.6 Å to the native heterodimer structure of H2A/H2B. Their structural alignments to the H2A/H2B native structure (Fig 5A) indicate that predictions from two totally different methodologies, AWSEM and AF2, agree well. We note that we have used two models of AF2-based prediction tools. Details of the two versions' performance and related discussion about protein complex prediction are in the Section S2 in S1 Text.

To examine the structural stability of the H2A/H2A histone homodimer, we further carried out explicit solvent all-atom simulations starting with the two initial conformations predicted by AWSEM, and AF2, respectively. We structurally aligned the simulation trajectory snapshots to the initial conformation, and calculated the root-mean-square-fluctuation (RMSF) of the $C_\alpha$ atoms to quantify local residual fluctuations (Fig 5B). The RMSF analysis displays comparable fluctuations between the two simulations on average within 3 Å. The RMSF analysis also shows that the loop regions of histone homodimer are in general more flexible than helical regions, as we expected. More RMSF analyses are in S11 Fig.

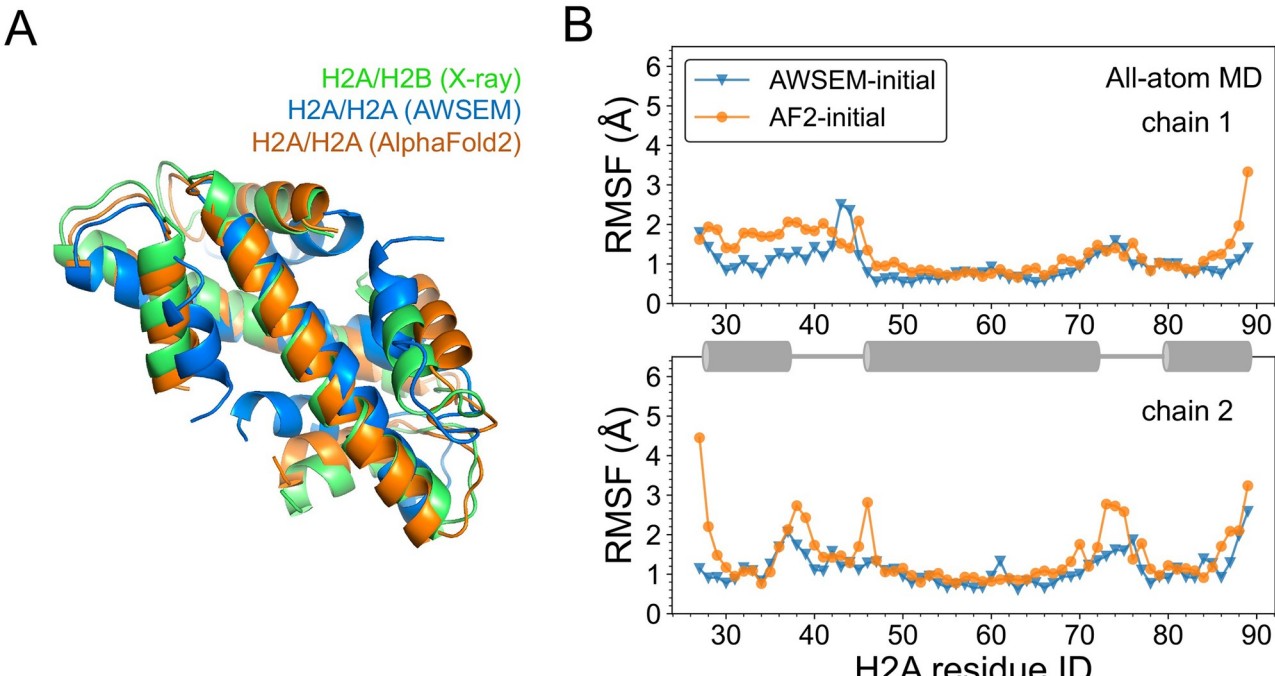

**Fig 5. Predicted structures of eukaryotic histone homodimer.** (A) AWSEM and AlphaFold2 predicted homodimer structures of H2A/H2A align well with the native heterodimer structure of H2A/H2B (colored in blue, orange, and green). (B) RMSF analysis of all-atom simulations demonstrates comparable stabilities of AWSEM- and AlphaFold2-predicted homo-complex structures (in blue and orange). RMSF of two chains are plotted separately and helix regions are animated by cartoon in grey.

## The role of histone tails in regulating dimer formation

The above results suggest that eukaryotic histone homodimers may be sufficiently stable in the absence of histone tails. This finding points to the possibility of histone tails tilting the balance between the homo- and heterodimer towards the latter. Hence, we next predicted the full-sequence histone homodimers using simulated annealing in AWSEM, and subsequently evaluated their structural similarities to the native heterodimer H2A/H2B (only for the histone fold core part) via the structural similarity measure, $Q_{dimer}$ (Fig 6A). Twenty independent simulated annealing runs were performed in AWSEM using the fragment memory of their homologs-allowed sequences (<95% sequence identity to target; details see Methods) for each system. It is evident that in most annealing runs, histones without tails form higher $Q_{dimer}$ compared to those with tails (Fig 6A), suggesting a significant inhibitory effect of histone tails in their dimer formation, whether in histone heterodimer or homodimer. In the presence of tails, the formation rate of histone heterodimers is notably higher than that of homodimers, corroborating our hypothesis that histone tails play a pivotal role in tipping the balance in favor of heterodimer formation. Representative structures show that full-sequence H2A/H2B form both native-like and inverted non-native histone-fold units (Fig 6B) while in H2A/H2A case, histone tails inhibit the formation of hand-shake motif (Fig 6C). Supplemental Q score analysis for the simulated annealing runs of four systems –H2A/H2A truncated and full-sequence, H2A/H2B truncated and full sequence– was provided in S12 Fig.

Moreover, we used AlphaFold-Multimer to predict the full-sequence structure of H2A/H2A. The obtained structure is very similar to that of H2A/H2B, with a $Q_{dimer}$ of 0.7 (Fig 6D). We then performed all-atom MD simulations by taking the predictions of AF-Multimer for

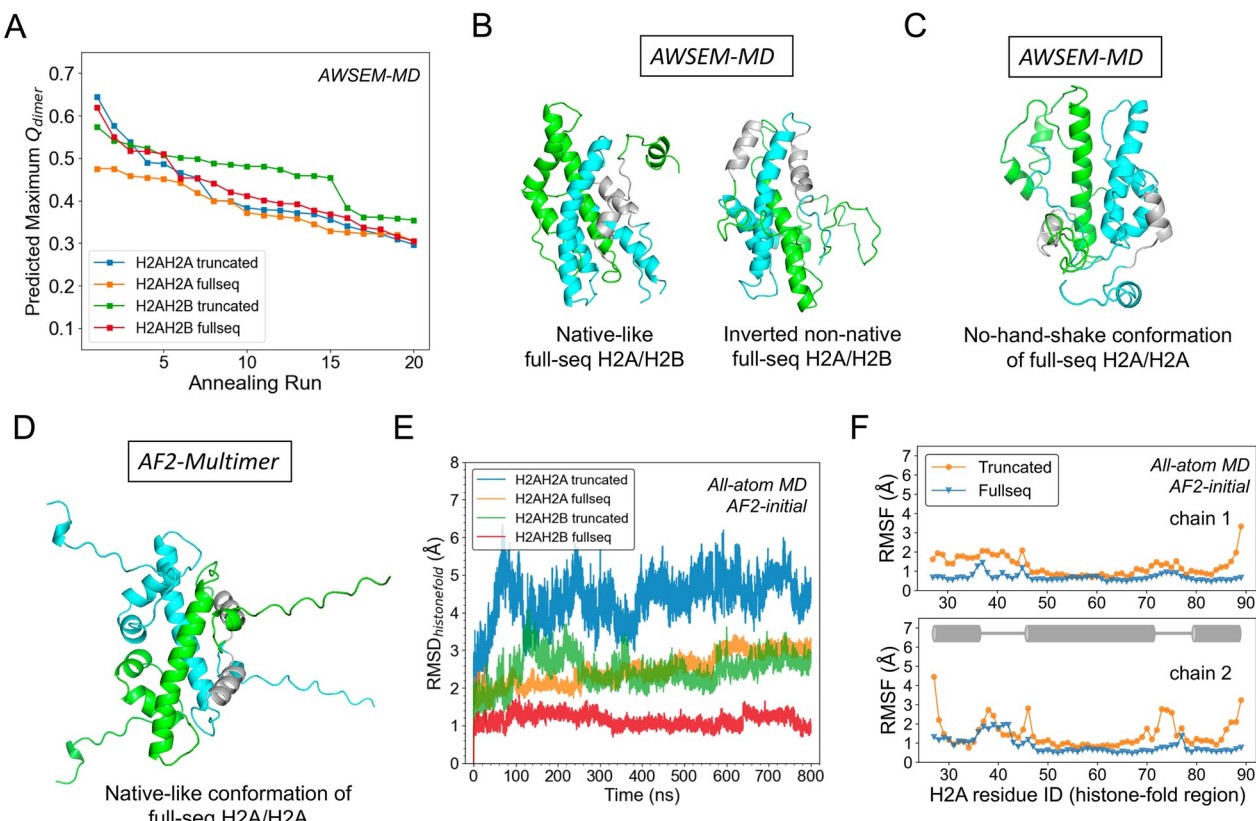

**Fig 6. Histone tails disfavor the formation of histone homodimer, yet stabilize the histone fold once formed.** (A) AWSEM-predicted trancated and full-sequence homodimer H2A/H2A are assessed by the maximum $Q_{dimer}$ of the histone-fold region (colored in blue and orange) and compared with predictions of trancated and full-sequence heterodimer H2A/H2B as a control (green and red). (B) AWSEM-predicted structures of full-sequence H2A/H2B (green/cyan) show native-like and inverted non-native orientations with displaced tail regions. (C) AWSEM-prediction of full-sequence H2A/H2A shows that the histone tails inhibit the formation of the hand-shake motif. $\alpha 1$ helices are colored in grey to help illustrate their native or non-native arrangements. (E) All-atom simulations of AlphaFold2-predicted truncated and full-sequence H2A/H2A homodimer (colored in blue and orange) are analyzed through RMSD and compared with that of truncated and full-sequence H2A/H2B (green and red). (F) The histone-fold region RMSF of the truncated and full-sequence H2A/H2A homodimer (colored in orange and blue). The two chains of H2A/H2A are plotted separately and their helix regions are animated by a cartoon diagram.

H2A/H2A homodimers, both with and without histone tails, as initial conformations and compared them with simulations of H2A/H2B in similar cases (full-sequence vs. truncated). The RMSD analyses confirmed that the predicted H2A/H2A structures, with truncated tails or not, are less stable than the corresponding H2A/H2B structures. Furthermore, interestingly, the full-sequence H2A/H2A and H2A/H2B have lower RMSD (histone-fold part), on average, than the truncated H2A/H2A or H2A/H2B (Fig 6E and S13 Fig). RMSF analysis (Fig 6F) shows that the flexibility difference between the truncated and full-sequence systems mainly appears at the terminal residues of the histone-fold core and the loop regions. More Alpha-Fold2-predicted H2A/H2A and H2B/H2B structures are provided in S14 and S15 Figs. Together, our simulations suggest that once the full-sequence H2A/H2A or H2A/H2B has formed a histone-fold core, the disordered tails may potentially help stabilize the loop regions.

## Discussion

Our study shows that the eukaryotic histone dimers not only have the same structural motif as the archaeal histones, but also share the same folding-upon-binding mechanism. The presence

of their binding partner may largely stabilize the mobility of folded secondary structures. Our further finding of energetically favorable native and inverted dimers points to the hidden ancestral symmetry of histone sequences. Not only are the conformation and sequence symmetry revealed here self-consistent, but more broadly, they are consistent with the symmetry-related folds, a recurring topic in protein folding studies [49–53]. Onuchic and colleagues studied the mirror image folds of three-helix bundle proteins [51]. Levy and Wolynes suggested a double-funneled energy landscape model to unravel the folding symmetry of a dimer protein [52]. Baker and colleagues developed a computational protocol to predict the structure of symmetrical protein assemblies and further a method to design protein homodimer based on the cyclic two-fold symmetry [50, 54]. In this context, the consistent symmetry between sequence and structure uncovered in this study may aid in unraveling the complexities of histone-like structures and functions, offering insights for applications in structural biology and protein design.

From a biology perspective, an interesting consequence of eukaryotic histones having forked from the archaeal tree is that eukaryotic histones have disordered tails and do not form homodimers. Our *in silico* results suggest that eukaryotic histones may form homodimers as well, with histone tails potentially destabilizing this formation. Positively charged histone tails are typically involved in binding to DNAs or various chromatin-modifying enzymes and transcription factors. Interestingly, H2B C-terminal tail carries two negative charges which may better favor the formation of H2A-H2B. However, a previous experimental work pointed that electrostatic repulsion in the N-terminal tails decreases the stability of the H2A-H2B dimer [44]. Future investigations could explore the influence of specific terminal tails of certain histones on the formation of homodimers or heterodimers. Besides the sequence and varying lengths of histone tails, their post-translational modifications may also affect the overall dynamics of histones and nucleosome [55–57]. Histone tails have undergone extensive investigation for their impact on nucleosome stability and binding with other chromosome regulator proteins [58–62]. Our study here delves into the folding dynamics of histones before they form nucleosomes. It is expected that their dynamics may differ once in a nucleosomal context.

The present discovery of the relatively stable inverted dimers and the inhibitory role of histone tails in the formation of their dimers suggests a general need for chaperones, whether proteins, DNA, or RNA, to facilitate the correct folding of histone dimers. This, in turn, ensures the proper binding for their subsequent interactions with DNA and other histone proteins. Indeed, a number of eukaryotic histone chaperones have been found in recent decades [63–68], such as Asf1 and MCM2 for H3/H4, Nap1 and MBD for H2A/H2B. Yet, most chaperones reported so far bind with dimeric histone complex. A recent exeperimental work by Pardal and Bowman reports that histone H3 and H4 can be separately chaperoned and transported by Imp5-H3 and Imp5-H4, and then get dimerized with other chaperones such as NASP and HAT1-RBBP7 [69]. This work suggests that the individual histone proteins H3 and H4 may be structurally stablized by Imp5. Besides, the revealed complex regulation network of histone chaperones can be a consequence of the typically existent conservation and symmetry that underly all histones and their binding partners both in sequence and structure. On this regard, we also anticipate the existence of archaeal histone chaperones or new chaperoning function from known proteins, which has not yet been reported.

However, we do not exclude the possibility that extreme ionic or temperature conditions which define the extremophile habitat, might somehow prevent the non-native association of archaeal histones. Furthermore, the ratio of native to non-native complexes varied among the different dimers that we simulated (S8 Fig). This implies that the sequence and structural diversities of histones, the varying lengths of histone tails, and post-translational modifications may all affect their overall folding dynamics. For instance, previously we reported that histone

monomers H3 and H4 play distinct structural and functional roles in their heterodimer H3/H4 complex [30]. It is possible that a chaperone would interact with only one histone monomer and primarily assist its folding. HJURP mainly interacting with CENP-A in the CENP-A/H4 dimer represents one such salient example [30]. Consequently, one may anticipate close co-evolution between histone monomers and chaperones.

## Conclusions

In this work, we used computational approaches supplemented with NMR and CD experiments to investigate the folding mechanism of histones. We found that a cooperative folding-upon-binding principle widely applies to canonical and variant eukaryotic histones, archaeal histones as well as histone-like transcription factors. Moreover, histone-fold structures may potentially form a low-energy non-native dimeric complex, in which on the tertiary structure level, an inverted arrangement is energetically nearly competitive with the native conformation. Subsequent sequence analysis revealed that this surprising non-native structure arises as a consequence of the ancient sequence symmetry underlying all histone proteins. Finally, we used AWSEM, AF-Multimer and all-atom MD simulations to predict possible formations of eukaryotic homodimers. Our results show that histone tails play an important role in regulating the formation of eukaryotic homodimer or heterodimer. Without tails, homotypic eukaryotic histones form potentially stable dimer in both AWSEM and AF-Multimer predictions, while with tails AWSEM simulations indicate an impeded formation of histone homodimers. Overall, our research illustrates common folding mechanisms of histone proteins, shedding light on their ancestral sequence and structural symmetries, and various new asymmetries engendered by evolution.

## Video available

Folding trajectories from AWSEM-MD simulations are provided respectively for the native and inverted non-native conformation of H2A/H2B (eukaryotic histones), and for the native and inverted non-native conformation of HmfA/HmfA (archaeal histones). Together, these videos highlight the symmetry and asymmetry behind histone folding across the eukaryotes and archaea.

## Supporting information

**S1 Text. Supplemental text for method detailss, extensive results and discussions.**
(PDF)

**S1 Video. Folding trajectories of eukaryotic histones and archaeal histones.** Folding trajectories of histones show the folding of the native and inverted non-native conformation of eukaryotic histones H2A/H2B and archaeal histones HmfA/HmfA.
(MP4)

**S1 Fig. Multiple sequence alignments (MSA) for all simulated proteins.** (A) MSA of the histone and histone-like proteins in this study show that the conserved residues (highlighted in black shade) are mostly hydrophobic and located near the dimer interfaces. (B) Full sequences of H2A and H2B aligned with histone-fold core of H2A and H2B, highlighting the extra length of histone tails on both N-terminal and C-terminal ends.
(TIF)

**S2 Fig. Simulations of archaeal histone monomer and histone-like protein monomer.** (A) Complementary to Fig 1B in the main context, here are simulation results for monomers of

archaeal histone and two types of transcription factor dTAF and NF-Y. (B) Structure alignments based on the longest $\alpha2$ helix (orange) for monomer simulation of H2A, H3, HMfA and NF-YC. Only the final conformations in corresponding annealing runs are included.
(TIF)

**S3 Fig. Clustering analyses for the final snapshots of histone monomer simulations.** Shown here are the clustering analysis results based on the pairwise RMSD among ten independent annealing runs for each histone monomer. No consensus structure was found.
(TIF)

**S4 Fig. Contact information of all the histone-like protein dimers in this study is accurately predicted by our simulations.** Contact maps are plotted for all the histone-like protein dimers in this study: (A) H2A/H2B; (B) H3/H4; (C) CENP-A/H4; (D) (HMfA)$_2$; (E) (HMfB)$_2$; (F) HMfA/HMfB; (G) dTAF$_{II}$42/dTAF$_{II}$62; (H) NF-YB/NF-YC dimer. Native and predicted contacts are represented in blue and orange, respectively. Predicted contacts for each dimer are computed using the structure of a simulation snapshot with the highest Q value from ten annealing runs.
(TIF)

**S5 Fig. Histone dimers and histone-like dimers fold upon binding.** Q values for the folded monomer and dimer are shown as functions of the annealing temperature in AWSEM-MD simulations of H3/H4 (A), CENP-A/H4 (B), (HMfA)$_2$ (C), (HMfB)$_2$ (D), and histone-like dimers dTAF$_{II}$42/dTAF$_{II}$62 (E), NF-YB/NF-YC (F). Markers and error bars represent the mean values and standard deviations of Q from ten independent simulation runs.
(TIF)

**S6 Fig. AWSEM-predicted archaeal histone heterodimer follows the same folding mechanism.** Previous biochemical studies demonstrate the formation of archaeal heterodimers HMfA/HMfB in vitro, which, however, was not structurally characterized. Based on our finding of eukaryotic histones (Fig 2), we hypothesize that archaeal histone heterodimers adopt a structure and binding-folding mechanism analogous to other eukaryotic histone dimers. We took HMfA and HMfB monomer structures from their homodimer complexes (PDB: 1B67 and 1A7W), and predicted their heterodimer structure using simulated annealing. We further applied umbrella sampling and calculated its free energy profiles. (A) Q value of the archaeal histone HMfA (blue) and HMfB (green), and its dimer (magenta) during the simulated annealing and the final prediction of archaeal heterodimer HMfA/HMfB structure are shown. (B) Free energy profile calculated at 300 K are projected on Q$_{monomer}$ and the $\alpha2$—$\alpha2$ angle (left), and radius of gyration Rg and the $\alpha2$—$\alpha2$ angle (right).
(TIF)

**S7 Fig. NMR experiments of H2B upon complex formation.** $^{15}$N{$^1$H} SOFAST-HMQC spectra of $^{15}$N-labeled H2B alone (A) and in the presence of unlabeled H2A at an equimolar ratio (B). (C) Heteronuclear steady-state $^{15}$N{$^1$H} NOE spectra recorded with amide proton presaturation for $^{15}$N-labeled H2B in the presence of unlabeled H2A at an equimolar ratio. In these spectra contours with positive intensities are colored black while negative intensities are blue.
(TIF)

**S8 Fig. Histone dimers and histone-like dimer proteins fold into native and non-native conformations.** Q$_{dimer}$ values are shown as functions of the annealing temperature for AWSEM-MD simulations of H2A/H2B (A), H3/H4 (B), CENP-A/H4 (C), (HMfA)$_2$ (D), (HMfB)$_2$ (E), HMfA/HMfB (F) and dimers dTAF$_{II}$42/dTAF$_{II}$62 (G), NF-YB/NF-YC (H). Ten individual simulation runs are represented in different colors and marker types. We categorize

all the runs with a final Q>0.5 as "native" and Q<0.5 as "non-native".
(TIF)

**S9 Fig. Potential energy from AWSEM-MD simulations shows both histone native and non-native conformations are energetically nearly degenerate.** The potential energy in AWSEM-MD includes the follow terms: $V_{backbone}$, $V_{contact}$, $V_{burial}$, $V_{HB}$, $V_{AM}$ and $V_{DSB}$ with details described in Davtyan *et al.* [23]. Numbers in this table are in the unit of *kcal/mol*.
(TIF)

**S10 Fig. Supplemental RMSD and representative snapshot of H2A/H2B conformation in all-atom simulations.** (A) Comparable stability of the inverted non-native conformation is consistently found to that of native structure in simulation replica 2. (B) The initial and final conformation (blue vs. orange) of the inverted structure display minor structural changes before and after 800-ns MD simulations.
(TIF)

**S11 Fig. Supplemental RMSF analyses of the homo-complex structures of H2A/H2A predicted by AWSEM and AlphaFold2 show similar structural flexibilities.** The RMSF analysis of all-atom simulations exhibit similar atomic flexibilities of AWSEM- and AlphaFold2-predicted homo-complex structures in different simulation replicas. The RMSF of two chains are plotted separately where their helix regions are schemed by cartoon in the middle.
(TIF)

**S12 Fig. Prediction of histone homodimer by AWSEM.** Q value of the histone-fold core region was calculated for the simulated annealing runs of four systems: H2AH2A truncated and full-sequence (colored in blue and orange), H2AH2B truncated and full sequence (green and read). The run that forms the highest $Q_{dimer}$ was chosen for each system. It shows that H2AH2B full sequence forms the most native-like dimer, more than H2AH2A truncated, H2AH2B truncated, and lastly the H2AH2A full sequence. Note that the formed Qdimer of H2AH2B is smaller than that in the dimer folding study as in Fig 2. This is due to the different fragment memory setup. Here, we want to "predict" homodimer structure of H2AH2A so the fragment memory excludes any homolog sequence with 95% identical to the target. To keep consistency, the same setup was used for the control group H2AH2B, which is different from the fragment memory setup in their dimer folding mechanism study.
(TIF)

**S13 Fig. Full-sequence histone homodimers show more stability than that of only histone-fold region.** (A) The histone-fold region in full-sequence homodimers has less RMSD (blue) than that of histone-fold only structures (orange) in different simulation runs. (B) The RMSF analyses in different runs demonstrate that the two ending sessions are particularly flexible in truncated homodimers.
(TIF)

**S14 Fig. Predicted structures of homo-complex of H2A/H2A by AWSEM and AlphaFold2.** Predicted structures of H2A/H2A by AWSEM (A) and AlphaFold2 (B) are shown, respectively. The two chains are colored in green and cyan while their $\alpha$1 helixes in grey to help illustrate their native or non-native arrangements.
(TIF)

**S15 Fig. AlphaFold2-predicted truncated and full-sequence homo-complex of H2B/H2B.** The two chains are colored by the same scheme which is the plDDT confidence score provided by AF2. Both truncated and full-sequence H2B/H2B are predicted to have a native-like

handshake structure while juxtaposing monomers are also found (full-sequence structure ii) which potentially indicates two non-interacting monomers.
(TIF)

## Acknowledgments

The authors thank Drs. Carlos Castañeda for help with initial NMR measurements, Tingting Yao for kindly providing plasmids for histone expression, and David Winogradoff, Guang Shi and Daniël Melters for helpful feedbacks.

## Author Contributions

**Conceptualization:** Haiqing Zhao, Yamini Dalal, David Fushman, Garegin A. Papoian.

**Data curation:** Haiqing Zhao, Hao Wu.

**Formal analysis:** Haiqing Zhao, Hao Wu, David Fushman, Garegin A. Papoian.

**Funding acquisition:** Haiqing Zhao, Yamini Dalal, David Fushman, Garegin A. Papoian.

**Investigation:** Haiqing Zhao, Hao Wu, Alex Guseman, Dulith Abeykoon, Christina M. Camara.

**Methodology:** Haiqing Zhao, Hao Wu, David Fushman, Garegin A. Papoian.

**Project administration:** Yamini Dalal, David Fushman, Garegin A. Papoian.

**Resources:** Alex Guseman, Dulith Abeykoon, Christina M. Camara.

**Software:** Haiqing Zhao, Hao Wu.

**Supervision:** Yamini Dalal, David Fushman, Garegin A. Papoian.

**Visualization:** Haiqing Zhao, Hao Wu, David Fushman.

**Writing – original draft:** Haiqing Zhao, Hao Wu, David Fushman.

**Writing – review & editing:** Haiqing Zhao, Yamini Dalal, David Fushman, Garegin A. Papoian.

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
