## [Decision Letter · Decision Letter 0]

18 Sep 2023

Dear Dr. Zhao,

Thank you very much for submitting your manuscript "The Role of Criptic Ancestral Symmetry In Histone Folding Mechanisms Across Eukarya and Archaea" for consideration at PLOS Computational Biology.

As with all papers reviewed by the journal, your manuscript was reviewed by members of the editorial board and by several independent reviewers. In light of the reviews (below this email), we would like to invite the resubmission of a significantly-revised version that takes into account the reviewers' comments.

While appreciating the significance of the study, both reviewer 2 and 3 are pointing out that the effect of histone tail on the stability of H2A/H2B dimer is not really explored in comparison to the sequence effect on inverted structure.

I hope that the authors can take advantage of this opportunity to fully address the reviewers' major and minor comments as well as clarify the technical issues.

As other minor points, 1. please fix a typography in the title (criptic -> cryptic), 2. Include "Author Summary", which is required in PLoS Comp. Biol.

We cannot make any decision about publication until we have seen the revised manuscript and your response to the reviewers' comments. Your revised manuscript is also likely to be sent to reviewers for further evaluation.

Sincerely,

Changbong Hyeon

Academic Editor

PLOS Computational Biology

Nir Ben-Tal

Section Editor

PLOS Computational Biology

While appreciating the significance of the study, both reviewer 2 and 3 are pointing out that the effect of histone tail on the stability of H2A/H2B dimer is not really explored in comparison to the sequence effect on inverted structure.

I hope that the authors can take advantage of this opportunity to fully address the reviewers' major and minor comments as well as clarify the technical issues.

As other minor points, 1. please fix a typography in the title (criptic -> cryptic), 2. Include "Author Summary", which is required in PLoS Comp. Biol.

Reviewer's Responses to Questions

**Comments to the Authors:**

Reviewer #1: In the present submission “The Role of Criptic Ancestral Symmetry In Histone Folding Mechanisms Across Eukarya and Archaea” H . Zhao et al. investigate homo- and heterodimeric folding for 11 different histone proteins. They apply a range of methods (coarse-grained and all-atom MD simulations, NMR and CD experiments) with a focus on theoretical/ simulation studies. They observe a structural competition of non-native symmetrically related dimeric folds (“native” vs. “inverted”) for tail-less histones.

Summed up, histone folding mechanisms are a topic of high biological interest and there is some discussion regarding their evolutionary changes. The present submission itself is well-written with a clear line of thought and investigate histone proteins in high detail. The technical aspects are well described. There are only few parts which need some minor work and should be addressed before publication (see below). After minor revision, the submission is suitable for publication.

Major Issues:

-

Minor issues:

-p. 5, methods and p.7 results: Any stochastic search for a global minimum can instead result in only identifying local approximations of the global minimum or even more dissimilar low energy structures. The author should include a short discussion how reliably using simulated annealing is when being applied on the AWSEM forcefield.

-p. 6 “Complete all-atom results are in S3.2”. should be replaced by something akin to “A detailed setup of the MD simulations is listed in in S3.2”.

-p. 10 “FE Landscape” might be replaced by “Free Energy Landscape” as it is a heading.

-p.13 Fig. 4a, the two dimeric chains could be colored differently for clarity.

-p.14 Last paragraph “in an unrelated protein, the Rop dimer, was…” Symmetrically related folds have been a recurring topic in the protein folding literature also for other cases than the Rop-dimer (e.g. mirror folds, J. Noel et al., J. Phys. Chem. B 2012, 116, 23, 6880–6888) and might be discussed in more broad terms.

Reviewer #2: This is an interesting study trying to shed some light on the evolution of structure and folding mechanisms of H2A/H2B dimer. In this respect, the title and abstract of the paper could be misleading and have to be renamed specific to H2A/H2B, other histone types have not been investigated. Overall paper is well written and has wonderful illustrations. The authors have observed the folding upon binding of H2A-H2B dimer in silico and experiments, this section is very solid. The finding of the inverted intermediate is also fascinating. I have several comments and suggestions which could approve the paper and make it clearer.

My main comment relates to the last result on histone tails’ contributions to the stabilization of the H2A/H2B heterodimer and destabilization of a homodimer. This finding and its discussion are rather vague. There could be several factors at play to explain this result:

- tails might not “tip the balance” if the homodimer is rearranged into “inverted” head-head conformations.

- As H2B tails contain negative charges unlike H2A tails, there is a possibility that tail-tail H2A-H2B interactions may stabilize the heterodimer.

- The authors applied AWSEM which uses the globular native protein fragments for biasing – were these fragments derived from heterodimers and histone octamers rather than from monomers, could it explain their results?

- Can the authors include plots for dimers with tails: Q vs annealing temperature?

- How relevant is this result to histones that function in the nucleosomal context? H2A/H2B histone tails in nucleosomes have different interaction and stabilization scope.

“A scaling analysis of the radius of gyration (Rg) as a function of the protein size suggests

that histone dimer acts as the minimum folded state, rather than its monomer” – the authors probably mean that histones form obligatory dimers?

“Our simulation indicates that histone monomers are highly disordered at the tertiary structure level” – they are mobile, but not disordered, at least not intrinsically disordered.

Figure 4 and Figure S2. Please say which program was used to align these sequences or structures.

Were the distance constraint applied for dimers and not for monomers? If yes, could it potentially bias the results on folding mechanisms?

How “completely unfolded state” was generated?

Folding of dimers is shown as a function of the annealing temperature in AWSEM-MD simulations – it would be beneficial (although probably time consuming so I do not insist on this) if the authors can apply their method to a dimer with monomers known to be structured in unbound state – for control.

Folding upon binding mechanisms, for some reason, are not even mentioned in Discussion.

Reviewer #3: The manuscript presents and interesting computational stud on the folding and binding mechanism of formation of histone proteins. The study is comprehensive in terms of the systems that are studied and the methods. It also combines some experimental results.

Comments

1. It seems that the presented study reports on two major results: the inverted symmetry due to the “quasi symmetric sequences” and its effect on the inverted structure and the effect of the tails. While the former is discussed in length, the latter is barely discussed. This asymmetry should be somehow addressed. There are no sequences of the tails, their length. No conformations of the tails and how they may interfere with folding. Also the references on the effect of disordered tails on folding or binding is limited.

2. The impression is that most of the results are stored in the SI. Particularly the results of some of the systems. The authors may consider to move some of the results to the main text.

3. Figure 4 is important but its caption is not clear enough. Particularly the relationship between (A) and (C)ii. Also shouldn’t some of the termini be labeled as C and N (without ‘)?

Also, it might be useful to show the location of the tails on these schemes?

4. Conclusions: One sentence says: “ without tails may form a low-energy non-native dimeric complex….” And another sentence “without tails, eukaryotic histones form potentially stable homodimers…”. One may read these as contradictory. A revision/clarification will be useful.

**Have the authors made all data and (if applicable) computational code underlying the findings in their manuscript fully available?**

Reviewer #1: Yes

Reviewer #2: **No: **I do not see a link to the data repository in the paper

Reviewer #3: Yes

PLOS authors have the option to publish the peer review history of their article (what does this mean?). If published, this will include your full peer review and any attached files.

Reviewer #1: No

Reviewer #2: No

Reviewer #3: No
---

## [Decision Letter · Decision Letter 1]

28 Nov 2023

Dear Dr. Zhao,

We are pleased to inform you that your manuscript 'The Role of Cryptic Ancestral Symmetry in Histone Folding Mechanisms Across Eukarya and Archaea' has been provisionally accepted for publication in PLOS Computational Biology.

Best regards,

Changbong Hyeon

Academic Editor

PLOS Computational Biology

Nir Ben-Tal

Section Editor

PLOS Computational Biology

Reviewer's Responses to Questions

**Comments to the Authors:**

Reviewer #2: The authors have addressed my comments.

**Have the authors made all data and (if applicable) computational code underlying the findings in their manuscript fully available?**

Reviewer #2: Yes

PLOS authors have the option to publish the peer review history of their article (what does this mean?). If published, this will include your full peer review and any attached files.

Reviewer #2: No

---

## [Editor Report · Acceptance letter]

29 Dec 2023

PCOMPBIOL-D-23-01309R1 

The Role of Cryptic Ancestral Symmetry in Histone Folding Mechanisms Across Eukarya and Archaea

Dear Dr Zhao,

I am pleased to inform you that your manuscript has been formally accepted for publication in PLOS Computational Biology. Your manuscript is now with our production department and you will be notified of the publication date in due course.

With kind regards,

Timea Kemeri-Szekernyes
